# Supramolecular catalyst with [FeCl$_4$] unit boosting photoelectrochemical seawater splitting via water nucleophilic attack pathway

Jiaming Miao[1], Cheng Lin[1], Xiaojia Yuan[1], Yang An[1], Yan Yang[1], Zhaosheng Li [2,3] ✉ & Kan Zhang [1] ✉

Propelled by the structure of water oxidation co-catalysts in natural photosynthesis, molecular co-catalysts have long been believed to possess the developable potential in artificial photosynthesis. However, the interfacial complexity between light absorber and molecular co-catalyst limits its structural stability and charge transfer efficiency. To overcome the challenge, a supramolecular scaffold with the [FeCl$_4$] catalytic units is reported, which undergo a water-nucleophilic attack of the water oxidation reaction, while the supramolecular matrix can be in-situ grown on the surface of photoelectrode through a simple chemical polymerization to be a strongly coupled interface. A well-defined BiVO$_4$ photoanode hybridized with [FeCl$_4$] units in polythiophene reaches 4.72 mA cm$^{-2}$ at 1.23 V$_{RHE}$, which also exhibits great stability for photoelectrochemical seawater splitting due to the restraint on chlorine evolution reaction by [FeCl$_4$] units and polythiophene. This work provides a novel solution to the challenge of the interface charge transfer of molecular co-catalyst hybridized photoelectrode.

Natural photosynthesis inspires us to utilize sunlight to address the energy and environmental crisis. In natural photosynthesis, the oxygen evolution reaction (OER) is the core reaction, in which the light absorber of chlorophyll is connected to the water oxidation catalyst with a metal center-ligand molecular structure, ensuring ideal charge transfer efficiency[1,2]. Enlightened by such delicate structures in nature, many efforts have focused on simulating those molecular structures to achieve the same complexity and precision of natural photosynthetic processes, such as metal-centric ligands or complexes, cubane complexes, polyoxometalate (POM), etc., which strive to reproduce the coordination environment, combined geometrical and electronic properties of natural water oxidation catalysts[3–5]. Until

now, molecular catalysts can achieve a high turnover frequency (TOF) of thousands for water oxidation that is much higher than inorganic cocatalysts[6–8]. Triggering considerable efforts devoted to anchoring the molecular catalysts onto light absorbers for building highly efficient artificial photosynthesis systems[8,9]. However, the solar-driven water oxidation efficiency in whatever artificial photosynthesis system is profoundly restricted by the interface between the light absorber and molecular catalyst. This limitation arises from the necessity of photo-induced charges to traverse across the interface, which is often hindered by unfavorable charge transfer abilities resulting from molecular anchoring. The anchor groups are always sensitive to the surrounding chemical environment, therefore

[1]School of Materials Science and Engineering, Nanjing University of Science and Technology, Nanjing 210094, China. [2]Jiangsu Key Laboratory for Nano Technology, Nanjing University, 22 Hankou Road, Nanjing 210093, China. [3]Collaborative Innovation Center of Advanced Microstructures, National Laboratory of Solid State Microstructures, College of Engineering and Applied Sciences, Nanjing University, 22 Hankou Road, Nanjing 210093, China. ✉e-mail: zsli@nju.edu.cn; zhangkan@njust.edu.cn

constraining the interfacial stability[10–12]. Therefore, solar-driven water oxidation based on molecular catalysts still faces a formidable challenge in their interface engineering.

Supramolecular catalysts, which are noncovalent crosslinks of linear polymers or macromolecular groups connected by non-covalent bonds, exhibit high and precise catalytic activity due to their free delocalized electrons and the presence of host-guest active coordination bonds[13,14]. More importantly, it can be further enhanced by incorporating π-stacking interactions and cross-links in the polymeric system to reinforce its structural rigidity[15,16]. However, the designed supramolecular catalysts have a complex structure and require meticulous synthesis procedures. Furthermore, although there are a few reports of supramolecular catalysts as OER active catalysts in photoelectrochemical (PEC) water oxidation, the crucial aspect of establishing a strongly coupled interface between supramolecular catalysts and light absorber has always been ignored, posing a great challenge to prolong its long-term stability[5,17,18].

In this work, aided by a $FeCl_3$ initiator, an $[FeCl_4]$ active unit in the supramolecular (polythiophene (PTh), polyaniline (PANI) or polypyrrole (PPy)) matrix can be naturally formed through a simple in-situ chemical polymerization strategy, which acts as an efficient OER catalyst to promote the charge carriers transfer, prolongs the charge lifetime and optimizes the surface reactions kinetics of a $BiVO_4$ (BVO) photoanode for PEC water splitting. While, the $[FeCl_4]$ active unit in the PTh presents the exceptional stability and anticorrosion even in the seawater, boosting photocurrent density of $BiVO_4$ photoanode from original 1.61 to 4.72 mA cm$^{-2}$ with long stability of 40 hours for PEC seawater splitting.

## Results

Porous BVO photoanode as a light absorber was synthesized based on Choi's method without further modification[19]. The supramolecular catalyst consisted of a PTh scaffold with $[FeCl_4]$ active units that can be in-situ grown on the surface of BVO by one-pot polymerization of thiophene monomers using $FeCl_3$ as an initiator (Fig. 1a). Briefly, $FeCl_3$ acts as an oxidant, which can oxidize the thiophene monomer to form cationic radicals. Then, these cationic radicals undergo spontaneous polymerization, resulting in the generation of elongated chains. The reduction of $FeCl_3$ to $FeCl_2$ occurs during the polymerization process, and a portion of $FeCl_3$ can trap chloride ions to form coordination bonds due to its high electron affinity and Lewis acid nature, resulting in a stable $[FeCl_4]$ tetrahedral structure formed in the PTh matrix[20–22].

The $[FeCl_4]$ unit is determined by $^{57}$Fe Mössbauer spectra. As revealed in Fig. 1b, the spectra can be well-fitted with two doublets whose Mössbauer parameters are collected in Table 1, corresponding to two coordination and doping modes that are similar to those of polythiophene and its derivatives[23–25]. Both principal doublets have the low isomer shift (IS) and quadrupole splitting (QS), which indicates that Fe is trivalent to form $[FeCl_4]$ units and the coordination number is six[26]. The lower quadrupole splitting represents an asymmetric distortion in the tetrahedral structure of $[FeCl_4]$ units, which provides the possibility of weak coordination with the S atom of polythiophene to form the S−$[FeCl_4]$−S, and this slight distortion will further improve the catalytic performance[27]. The structure of $[FeCl_4]$ is chemically stable, and its non-covalent coordination and interaction with polythiophene constitute a homogeneous iron-based supramolecular polymer. Furthermore, the Fe$^{III}$ in polythiophene has a larger density of

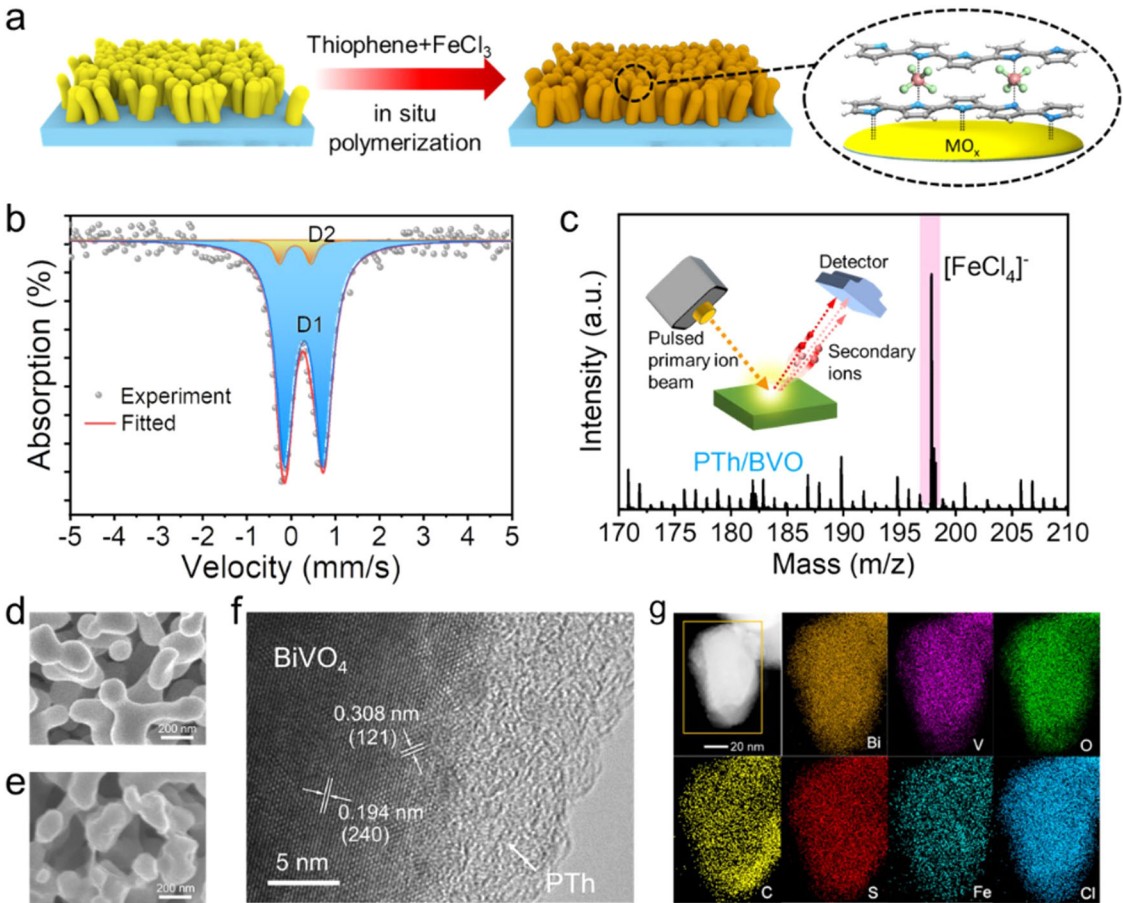

**Fig. 1 | Fabrication and characterizations of PTh/BVO photoanode. a** Electrode preparation process by in-situ polymerization of polythiophene. **b** Mössbauer spectra of PTh/BVO. **c** TOF-SIMS spectra of PTh/BVO. **d** SEM image of BVO. **e** SEM image of PTh/BVO. **f** HRTEM images of PTh/BVO. **g** HAADF-STEM-EDX elemental mapping of PTh/BVO.

**Table 1 | Mössbauer parameters of PTh/BVO**

|  | IS (mm/s) | QS (mm/s) | Area (%) |
|---|---|---|---|
| Doublet 1 | 0.30 | 0.87 | 94.1 |
| Doublet 2 | 0.10 | 0.71 | 5.9 |

3d electrons and it performs a high spin state, which would be beneficial for the OER process[28,29]. The [FeCl₄] unit in the supramolecular structure of polythiophene is further demonstrated by time-of-flight secondary ion mass spectrometry (TOF-SIMS) measurement (Fig. 1c and Supplementary Fig. 1)[30]. Different kinds of secondary ions fly at different speeds due to specific mass charge ratios and thus can be detected sensitively. The typical secondary negative-ion $[FeCl_4]^-$ is detected at 197.8 m/z, additionally, typical ion fragment signals attributed to BVO electrodes ($VO_3^-$, 98.9 m/z) and thiophene ($C_4HS^-$, 81.0 m/z) are also detected, which further demonstrate the existence of a Fe-based coordination structure.

The composition and electronic states of BVO and PTh/BVO photoelectrodes are compared by X-ray photoelectron spectroscopy (XPS) and Fourier transform infrared spectroscopy (FTIR) spectra (Supplementary Fig. 2-6). As shown in Supplementary Fig. 2, accessorial signals of Fe, S and Cl can be found following the in-situ growth of polythiophene. In high-resolution XPS spectra of PTh/BVO, the Cl 2p signal attributed to the [FeCl₄] in the polythiophene can be observed at a binding energy of 198.0 eV and 199.6 eV, while accompanying a PTh−Cl signal at 200.1 eV caused by the charge-transfer interaction between Cl ions (introduced by $FeCl_3$) and PTh (Supplementary Fig. 3a). In S 2p XPS spectra of PTh and PTh/BVO (Supplementary Fig. 3b), in addition to verifying the existence of [FeCl₄] unit in the polythiophene, a new S 2p peak appears at 163.5 eV for PTh/BVO, and all of S peaks shift towards high energies, which is attributed to the nucleophilic S atom bonding to the empty orbital formed by the unpaired lone pair electrons of Bi atoms at the BVO surface. The formed chemical bonding is believed to facilitate the charge transfer and reinforce the interface stabilization[31,32]. Similarly, a shift towards higher energies is also observed in Bi 4 f and V 2p XPS spectra (Supplementary Fig. 4), suggesting that the change in electronic structure can be attributed to the interaction between Bi atoms and the polythiophene layer at the interface region, with polythiophene serving as an electron acceptor[33]. Fe 2p XPS spectra in Supplementary Fig. 5 clearly show Fe signals in both PTh and PTh/BVO, with the identification of four distinct peaks. The binding energy and signal parameters of Fe 2p XPS spectra differ from those of previously reported Fe-based oxides and hydroxide cocatalysts, which indicates a coordination compound nature of Fe−Cl in polythiophene without any other impurities[34−36].

Scanning electron microscopy (SEM) and transmission electron microscopy (TEM) images of BVO and PTh/BVO photoelectrodes are shown in Fig. 1d-e. The pristine BVO exhibits a three-dimensional porous structure with an average grain size of more than 100 nm (Fig. 1d) and the thickness of the film is about 1.2 μm (Supplementary Fig. 7). After the in-situ polymerization of PTh on the surface of BVO, a roughened morphology is observed (Fig. 2e). X-ray diffraction (XRD) patterns of BVO and PTh/BVO photoelectrodes exhibit no significant distinction (Supplementary Fig. 8), with all diffraction peaks can be indexed to the monoclinic BVO (JCPDS: 14-0688). High-resolution TEM (HR-TEM) image reveals that an amorphous PTh layer with a thickness of around 10 nm is tightly and uniformly coated on the BVO surface (Fig. 2f). Moreover, the lattice fringes of BVO are clearly discernible with a spacing of 0.308 nm, which corresponds to the (121) plane of monoclinic BVO, while the lattice spacing of 0.194 nm can be indexed to the (240) plane of monoclinic BVO. Energy-dispersive X-ray spectroscopy (EDS) mapping of PTh/BVO shows that the signals of Bi, V, O, C, S, Fe and Cl elements are uniformly distributed in the entire region (Fig. 1g).

Due to the chemically bonded interface formed between the polythiophene and BVO, the transfer of photogenerated charge carriers at the interface would be significantly promoted. The kinetics of surface charge transfer and recombination characteristics were investigated by intensity-modulated photocurrent spectroscopy (IMPS) spectra (Fig. 2a-d). In the IMPS spectra, a high-frequency semicircle located in quadrant IV represents bulk charge diffusion, while a low-frequency semicircle located in quadrant I corresponds to the surface-associated process, respectively[37]. The rate of charge transfer and recombination can be represented by two constants $k_{trans}$ and $k_{rec}$. In the normalized IMPS spectrum, $k_{trans} / (k_{trans} + k_{rec})$ can be obtained from the intercept of the low-frequency semicircle, and $k_{trans}+k_{rec}$ can be determined from the frequency at the maximum imaginary current[38,39]. According to Fig. 2a-b, in the bias range of 0.4-1.2 V vs. RHE, the IMPS plots of PTh/BVO have a smaller radius of the high-frequency semicircle and a larger intercept with x-axis than those of BVO. The aforementioned finding provides evidence for the enhanced charge transfer efficiency and favorable carrier kinetics of the PTh/BVO photoanode. In addition, the pseudo-first-order rate constants $k_{trans}$ and $k_{rec}$ can quantitatively describe the rates of charge transfer and recombination. As shown in Fig. 2c, the PTh/BVO photoanode exhibits lower $k_{rec}$ in the bias range, implying that surface charge recombination is well suppressed. Moreover, the PTh/BVO photoanode also demonstrates a higher transfer rate constant $k_{trans}$ than the BVO photoanode, which plays a determining role in the surface reaction kinetics (Fig. 2d)[40]. These characteristics are consistent with those molecular catalysts loaded cases[17,41], which confirm that the supramolecular PTh layer with [FeCl₄] unit can act as an efficient OER catalyst to not only accelerate the kinetics of water oxidation but also alleviate surface charge recombination.

To further disclose the charge carrier kinetics process, time-resolved photoluminescence (TRPL) and transient-state surface photovoltage (TS-SPV) spectra were measured (Supplementary Fig. 9). As shown in Fig. 2e, fitting the TRPL spectra with a bi-exponential function reveals two distinct decay components contributing to the charge carrier lifetime: a fast decay component ($\tau_1$), attributed to intrinsic defect trapping of photo-generated charges, and a slow decay component ($\tau_2$), associated with electron-hole recombination processes[42−44]. All parameters of the TRPL measurement are summarized in Table 2.

It can be seen that both $\tau_1$ and $\tau_2$ of PTh/BVO are longer than that of BVO. In particular, $\tau_2$ of PTh/BVO elevates to 7.30 ns compared to pristine BVO (4.65 ns), which proves that the enhanced separation and diffusion of electron-hole pairs due to the inhibited surface recombination by the PTh layer[45]. Compared to TRPL spectra, the TS-SPV spectra can reflect the generation, separation, migration and recombination of charge carriers on the electrode surface, and directly exhibit the photogenerated charge concentration and lifetime. The TS-SPV measurements were carried out under 355 nm laser radiation, and the results are presented on a normal timescale (Supplementary Fig. 10) and a logarithmic time scale (Fig. 2f), respectively. It can be seen that the recombination time of the PTh/BVO photoanode is approximately 10 times longer than that of the BVO photoanode. Moreover, the signal intensity of PTh/BVO is more than 42 times stronger than that of BVO. Interestingly, the time domain of the TS-SPV spectrum of PTh/BVO exhibits distinguishable characteristics, from which behaviour of charge carrier kinetics can be divided: 1) when the laser radiates, photogenerated holes of BVO are injected into the PTh layer because of the built-in electric field at the interface and the holes extraction by [FeCl₄] unit, this process is quite rapid to reach the maximum and is manifested by a sharp signal response peak ($1.2 \times 10^{-7}$ s, Peak 1)[46]; while, the photoexcited electrons of supramolecular catalyst in PTh layer are also injected into the BVO, whereas the latter process is negligible; 2) a minority of the surface holes are trapped by

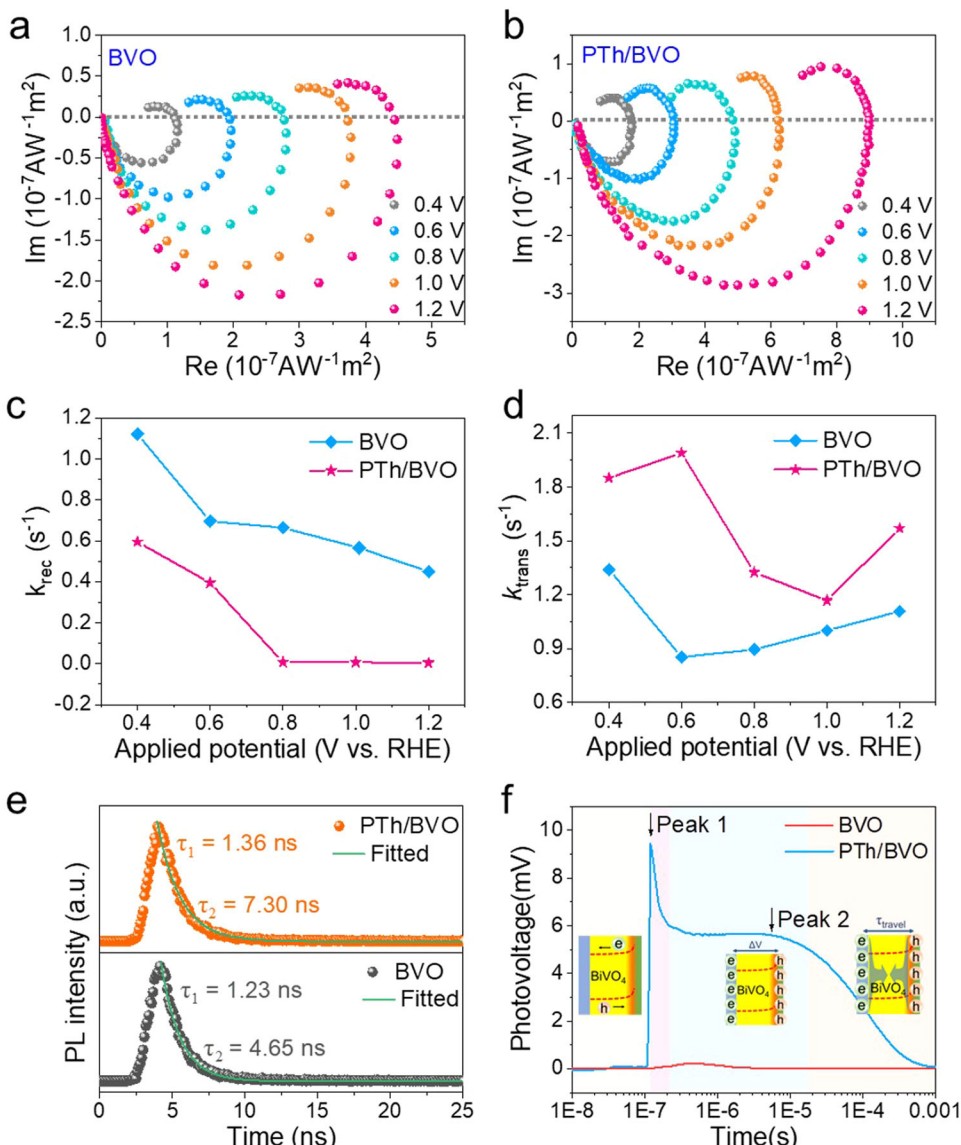

**Fig. 2 | Charge carrier kinetics of PTh/BVO photoanode.** MPS plots of (**a**) BVO and (**b**) PTh/BVO. **c** Recombination rate constant ($k_{rec}$) and **d** charge transfer rate ($k_{trans}$). **e** Time-resolved photoluminescence (TR-PL) spectra. **f** Transient-state SPV measurements under 355 nm laser irradiation on a logarithmic timescale of BVO and PTh/BVO.

the surface states, so the photovoltage signal has a small period of sharp drop ($1.2\times10^{-7}$ to $4.0\times10^{-7}$ s);[47] 3) the Fe sites located in the PTh matrix stores the injected holes by their changeable valence, and built-in electric field mitigates the recombination of charge carriers, so that these holes have long lifetime and avoid fast charge recombination, which is displayed as a flattening of the photovoltage signal and reaching a stable intensity ($2.0\times10^{-6}$ s, Peak 2); 4) eventually, the photogenerated holes and electrons are completely recombined. The results illustrate that while not impeding charge transport properties, the presence of PTh layer enhances additional charge extraction and storage capacity, thereby improving the interface effect between the light absorber and molecular catalyst.

## Table 2 | Biexponential decay-fitted parameters of TRPL for BVO and PTh/BVO photoanodes

| Samples | $\tau_1$ (ns) | $\tau_2$ (ns) | $\tau_{avg}$ (ns) | $A_1$ | $A_2$ | $\chi^2$ |
|---|---|---|---|---|---|---|
| BVO | 1.23 | 4.65 | 1.40 | 551.72 | 14.10 | 0.521 |
| PTh/BVO | 1.37 | 7.30 | 1.71 | 577.87 | 6.51 | 0.609 |

To demonstrate the [FeCl$_4$] unit on improving the PEC performance, the PTh layers on the BVO surface were polymerized by other two initiators, (NH$_4$)$_2$S$_2$O$_8$ (APS) and HAuCl$_4$. PEC performances of BVO and three kinds of PTh/BVO (noted PTh-Fe/BVO, PTh-APS/BVO and PTh-Au/BVO) photoanodes were measured in 0.5 M borate buffer (pH = 9.0) electrolyte under AM 1.5 G simulated sunlight illumination (100 mWcm$^{-2}$). It can be seen that the photocurrent densities of three kinds of PTh/BVO photoanodes are all higher than that of BVO, indicating the positive effect of the PTh layer on improving charge carrier kinetics. Remarkably, the PTh/BVO with FeCl$_3$ as initiator has much higher photocurrent density than those with APS and HAuCl$_4$ as initiators. By regulating the concentration of FeCl$_3$, polymerization time and electrolyte, the precise utilization yields of FeCl$_3$ and the optimal experimental parameters were obtained (Supplementary Fig. 11). The photocurrent density increases from 1.61 mA cm$^{-2}$ for BVO to 4.72 mA cm$^{-2}$ for PTh/BVO with FeCl$_3$ as initiator at 1.23 V vs. RHE, which is approximately 2.9 times higher than the pristine BVO photoanode and also stands out among the previous counterparts (Fig. 3a, Supplementary Table 1). Furthermore, an evident cathodic onset potential shift ($\sim$ 220 mV) is observed, indicating robust charge

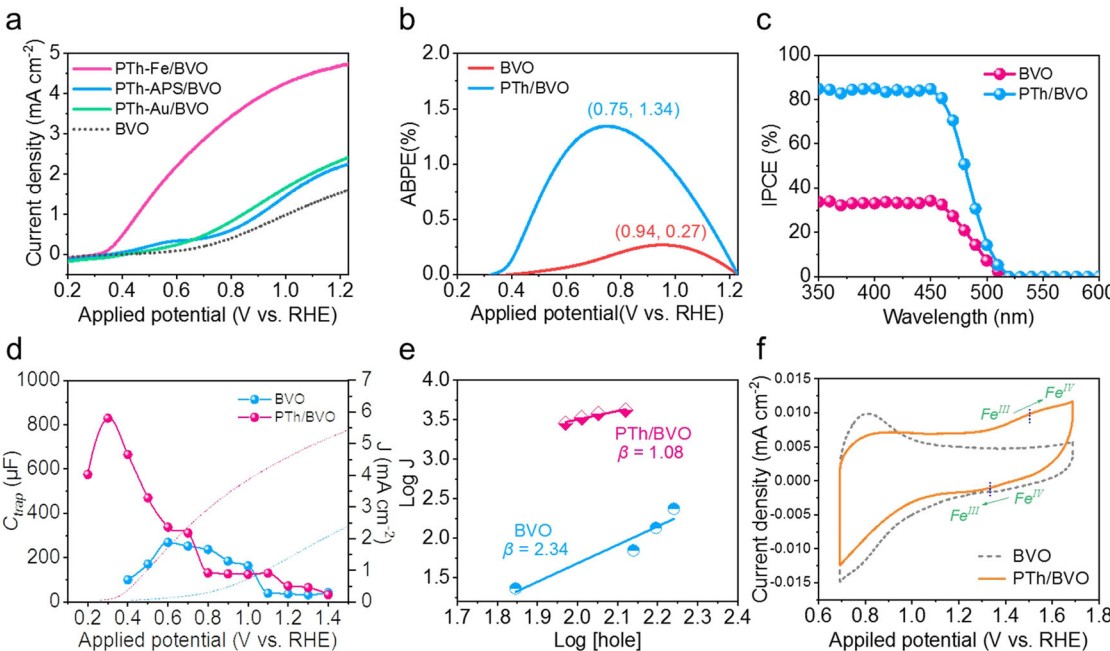

**Fig. 3 | Photoelectrochemical properties of PTh/BVO photoanode. a** LSV curves of BVO and PTh/BVO with 0.5 M KBi (pH = 9) under AM 1.5 G (100 mW cm$^{-2}$) illumination. **b** ABPE. **c** IPCE. **d** J–V curves (dashed lines) and $C_{trap}$ values (solid lines and dots). **e** Relationship between the photocurrent density and surface-hole density. **f** CV curves of BVO and PTh/BVO.

separation efficiency and optimized reaction kinetics[48]. Moreover, the OER activities of the PTh/BVO with different initiators are also elucidated in Supplementary Fig. 12. A great contribution from [FeCl$_4$] unit to OER activity enables the cathodically shifted overpotential by -297 mV compared with that of BVO (Supplementary Fig. 13a), and the Tafel slope is decreased from 386 mV/decade of BVO to 264 mV/decade of PTh/BVO (Supplementary Fig. 13b). The results demonstrate that the PTh supramolecular with [FeCl$_4$] unit could serve as a co-catalyst to improve OER reaction kinetics, as well as facilitate charge transfer process[49]. Additionally, the chopped transient photocurrent curve at 1.23 V vs. RHE is shown in Supplementary Fig. 14. The presence of severe electron-hole recombination at the interface will cause a steep spike of the photocurrent; however, the PTh/BVO shows a negligible spike, ascribing to fast water oxidation kinetics[50]. Correspondingly, the charge transfer efficiency ($\eta_{transfer}$) and charge transport efficiency ($\eta_{transport}$) are calculated by using sodium sulphite as a hole scavenger (Supplementary Fig. 15-17). Notably, the PTh/BVO photoanode exhibits a higher $\eta_{transfer}$ value (81.3%) of PTh/BVO than that of BVO photoanode (36.6%) at 1.23 V vs. RHE, but a little change in $\eta_{transport}$ due to the primary co-catalyst effect on OER reaction kinetics. Applied bias photon-to-current efficiency (ABPE) of BVO and PTh/BVO is calculated in Fig. 3b. The ABPE for PTh/BVO is 1.34% at 0.75 V vs. RHE, which is much higher than that of pristine BVO (0.27%, 0.94 V vs. RHE). The favorable photoconversion efficiency at low potentials will be the most significant feature, which holds great promise for achieving unbiased solar water splitting. Incident photon to current conversion efficiencies (IPCE) of BVO and PTh/BVO photoanodes are shown in Fig. 3c. The IPCE value after light harvesting efficiency correction of PTh/BVO reaches above 80% at wavelengths from 350–450 nm at 1.23 V$_{RHE}$, which is much higher compared with pristine BVO. The UV spectra (Supplementary Fig. 18) show that there is negligible enhancement of light absorption, thus emphasizing the pivotal role of charge transfer in the elevation of IPCE.

To gain further insights into the role of the PTh matrix with [FeCl$_4$] unit as a highly efficient co-catalyst for OER, Mott-Schottky (M-S) and electrochemical impedance spectroscopy (EIS) measurements were investigated. As displayed in Supplementary Fig. 19, the flat band potential ($E_{fb}$) of BVO, which is determined by extrapolating the Mott–Schottky plot to $1/C^2 = 0$, positively shifts from 0.38 to 0.49 V after PTh coating. This change in $E_{fb}$ indicates the formation of a p-n junction[51]. In addition, according to the effective surface area of the porous BVO electrode, the charge carrier densities ($N_D$) of PTh/BVO are calculated as $3.96\times10^{19}$ cm$^{-3}$, which is 5.8 times higher than that of BVO ($6.87\times10^{18}$ cm$^{-3}$), corroborating that [FeCl$_4$] unit robust the charge transport process. Furthermore, the equivalent circuit for interpretation of the EIS results is illustrated in Supplementary Fig. 20 and the electrochemically active surface areas (ECSA) are shown in Supplementary Fig. 21[52]. The higher slope of PTh/BVO illustrates that the redox-active Fe sites have increased the OER active surface, which further implies the enhancive double-layer capacitance. The fitted capacitance of the surface state ($C_{trap}$), as well as the photocurrent density for BVO and PTh/BVO are illustrated in Fig. 3d. The $C_{trap}$ values of both electrodes reach the highest at the photocurrent onset potential, while the quickly decreasing $C_{trap}$ value of PTh/BVO indicates a rapid depletion of surface holes during water oxidation. Further on, the PTh/BVO exhibits a significant surface charge release rate, and the accumulated charges are completely depleted within a narrow voltage range, which proves the faster water oxidation process on the surface of PTh/BVO accounting for the enhanced photocurrent density[53,54]. Supplementary Fig. 22 exhibits the illumination intensity modulated EIS spectra of BVO and PTh/BVO. The reaction order can be calculated from log/log plots of photocurrent density and hole density to further explain the OER dynamics and mechanism on the surface. As shown in Fig. 3e, the reaction orders of PTh/BVO and BVO are 1.08 and 2.34, respectively. According to previous reports, the results suggest that the OER on the surface of PTh/BVO is mainly in accordance with the water nucleophilic attack (WNA) mechanism since [FeCl$_4$] units act as the active sites of OER like the coordination metal centers in molecular catalysts[11,55], and this one-hole transfer pathway is easier to be realized, thus allowing the surface charge to participate in the water oxidation reaction more rapidly[56]. Subsequently, kinetic isotope effects (KIE) are used to deeply explicate the mechanism of water oxidation, which contains information about proton transfer

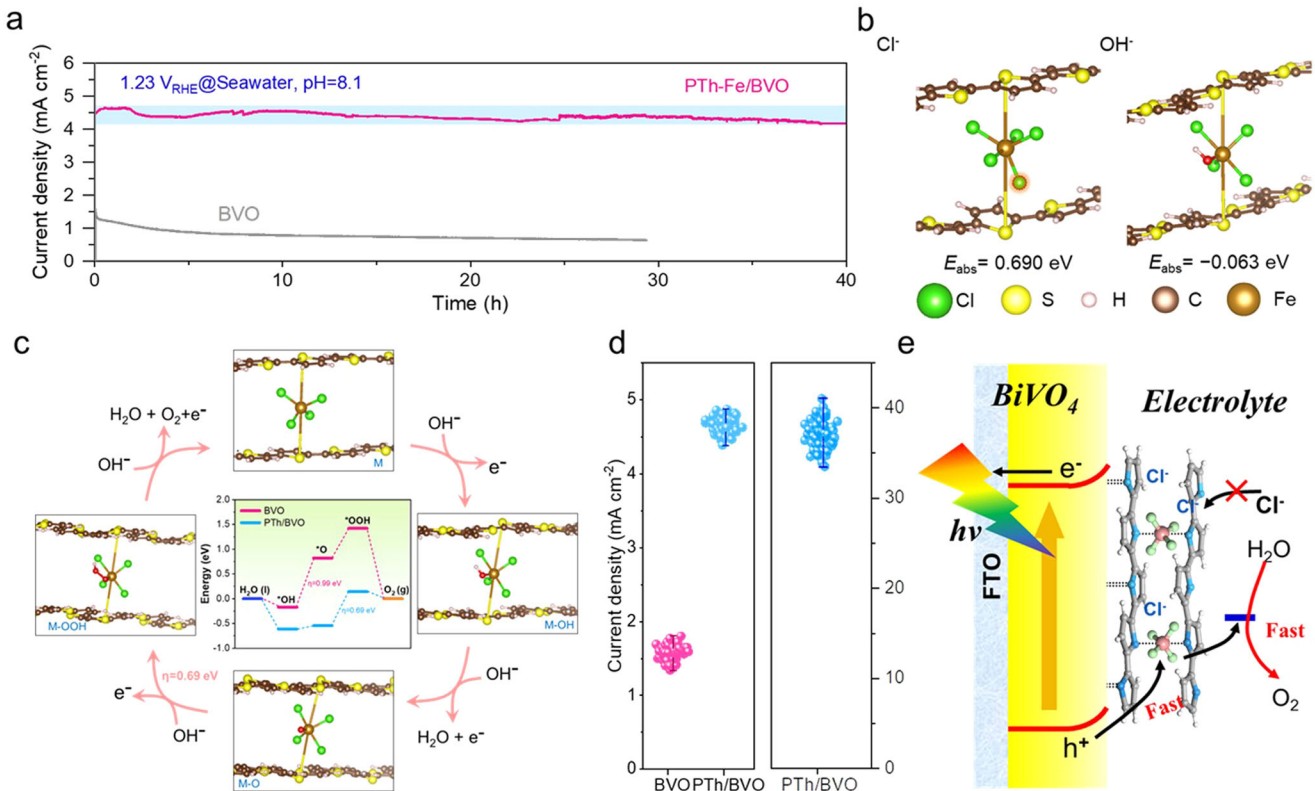

**Fig. 4 | Photoelectrochemical performance for seawater splitting. a** Long-term PEC stability of PTh/BVO and BVO at 1.23 V vs. RHE in seawater electrolyte (pH = 8.1) under AM 1.5 G illumination. **b** Atomic structures of $Cl^-$ and $OH^-$ adsorption on [FeCl$_4$] unit. **c** Schematics and free energy diagram for OER pathway on PTh/BVO. **d** Scatter plots of photocurrent densities of PTh/BVO and BVO photoanodes. **e** Schematic illustration of the charge transfer process in PTh/BVO.

kinetics, so as to reflect the rate-determining step (RDS) in the whole reaction process. The KIE values are defined as the ratio of the photocurrent densities measured in water and deuteroxide electrolytes (Supplementary Fig. 23). It can be observed from Supplementary Fig. 23b that the KIE values of BVO are all higher than 2, indicating that the proton transfer process is the RDS of the water oxidation reaction, corresponding to the O−H bond rupture[17]. The KIE values of PTh/BVO are less than that of BVO, which means that the proton transfer process is well promoted, thus the water oxidation reaction kinetics can be improved more effectively and holes can be fast transferred to the water[57,58]. This can be elucidated by the good hydrophilicity of PTh (Supplementary Fig. 24), as well as the reduced energy barrier of water oxidation intermediates formation.

To further clarify the effect of the [FeCl$_4$] active site presented in the PTh layer on the OER process, cyclic voltammetry (CV) curves are shown in Fig. 3e. From the CV curves, it is clear that there are two oxidation peaks before the onset potential of steady-state water oxidation. Compared with the CV curve of BVO, the peak at 0.81 V may be attributed to the oxidation of species V, and this oxidation can be analyzed to be inhibited by the PTh layer[59]. A slight peak located at 1.54 V can be ascribed to the oxidation of the Fe site, indicating that a small amount of Fe in the PTh layer undergoes a Fe$^{III}$ to Fe$^{IV}$ transition, and this valence change will cause a part of the holes to be stored[60]. The hole storage capacity of PTh can facilitate the efficient participation of holes in the water oxidation process at the electrode/electrolyte interface[61]. On the other hand, these Fe species in the PTh matrix acted as redox-active sites, and the turnover frequencies (TOF) versus overpotential were calculated in Supplementary Fig. 25.

Stability and reproducibility are two important properties that determine the generalizability of photoanodes for practical applications. The i-t curve in Supplementary Fig. 26 indicates that the PTh/BVO has stability at least for 50 hours. To our surprise, PTh/BVO also

has high photocurrent density (Supplementary Fig. 27), fast kinetics of photo-excited charge (Supplementary Fig. 28), and great stability in the seawater electrolytes. The pH and main components of the seawater used in this work are determined in Supplementary Table 2. As shown in Fig. 4a, the photocurrent density of the PTh/BVO can remain well stable for at least 40 hours for PEC seawater splitting. In sharp contrast, the BVO is rapidly corroded, which is probably due to chlorine evolution reaction (CER) that accelerates the photocorrosion effect. To understand the CER inhibition via the PTh/BVO, the amounts of $H_2$ and $O_2$ products during PEC seawater splitting of 10 hours were measured, which are approximately 795.8 and 369.2 μmol, respectively, (Supplementary Fig. 29). The Faradaic efficiency of oxygen evolution reaction is calculated to be 91.7%, which is comparable value to previous reports (Supplementary Table 3). The ICP-MS measurements further reveal that the dissolution of V element is significantly reduced compared to the bare BVO photoanode after 40 hours of i-t measurement (Supplementary Fig. 30), and the morphology and chemical structure of the photoanode do not change significantly (Supplementary Fig. 31-35). Different from the previous surface functionalization and passivation strategies, this phenomenon can be attributed to the $Cl^-$-storage properties of polythiophene. As shown in Supplementary Fig. 36, in addition to the oxidation peak of V species, the CV curve of PTh/BVO electrode shows an oxidation peak and a reduction peak at 0.66 and 0.36 V vs. RHE, respectively. The redox peak can be associated with the electrochemical reactions of Cl storage in PTh, as expressed in Eqs. (1) and (2):

$$PTh + Cl^- \rightarrow PThCl + e^- \qquad (1)$$

$$PThCl + e^- \rightarrow PTh + Cl^- \qquad (2)$$

To determine the Cl⁻ storage properties of polythiophene, the Cl 2p XPS spectra after long-term stability testing were investigated. As shown in Supplementary Fig. 37, the peak at 200.1 eV associated with the stored Cl ions is ascribed to the charge-transfer interaction between the Cl ions and the polythiophene chain. The disappearance of such a peak is observed after i-t testing in the KBi solution, while observing a slight increase in peak intensity after i-t testing in the seawater. This ox-red cycling of Cl ions could avoid direct contact of Cl⁻ ions with the BVO surface. Moreover, the inhibitory effect on chlorine oxidation by Fe−Cl coordination bonding exists in the PTh matrix[62]. As shown in Fig. 4b, the adsorption energies of Cl⁻ and OH⁻ on [FeCl₄] unit are theoretically calculated. The adsorption energy of Cl⁻ is 0.690 eV, indicating an endothermic process with limited occurrence, while the adsorption energy of OH⁻ is −0.063 eV, suggesting a higher propensity for spontaneous adsorption. Due to the higher adsorption capacity for OH⁻, the WNA mechanism occurs more tendentiously during the water oxidation, in which OH⁻ continuously nucleophilic attacks to complete the proton-electron coupling process. The energy barriers of water oxidation for BVO and PTh/BVO are established via density functional theory (DFT) calculations (Fig. 4c). It can be seen that the second step (*OH → *O) is the rate-determining step for BVO with about 0.99 eV overpotential. For PTh/BVO, both the OH* and O* steps exhibit low energy barriers, and the lower overpotential with 0.69 eV of the rate-determining step (O* → OOH*) indicates the optimized water oxidation kinetics.

Furthermore, we fabricated 100 PTh/BVO photoanodes (Fig. 4d), their photocurrent densities and stability present a fluctuation of only 2.55% and 4.40%, indicating good reproducibility. Based on this, we scaled up this strategy to fabricate a large area PTh/BVO photoanode with 5×6 cm², leading to similar PEC performance (Supplementary Fig. 38). The schematic diagrams of the PTh/BVO for PEC water splitting are shown in Fig. 4e. Profited by supramolecular like PTh/[FeCl₄] catalyst modification, the sturdy "S−Bi" bonding enables the interface stable enough to optimize the charge transfer and the lifetime of the photogenerated hole. The Cl⁻ stored in the PTh matrix can effectively avoid the corrosion of CER to the electrode, enabling molecular catalyst-like behaviors but superior stability to molecular catalysts. In addition to the PTh, this supramolecular co-catalyst can also be extended to other conductive polymers, such as polyaniline (PANI) and polypyrrole (PPy)[56]. By using FeCl₃ as an initiator, the [FeCl₄] active sites in PANI and PPy are still similar to PTh (Supplementary Fig. 39), by which both PANI/BVO and PPy/BVO exhibit similar morphology (Supplementary Fig. 40) and positive effect on improving PEC performance and stabilities in the seawater (Supplementary Fig. 41-42). Furthermore, we applied this supramolecular polymer strategy to TiO₂ and WO₃ photoanodes, the results show a similar PEC performance improvement (Supplementary Fig. 43-44).

## Discussion

In summary, we have reported a strong interface coupled polymeric PTh, PANI and PPy/BVO photoanode via a simple in-situ FeCl₃ initiator-induced polymerization strategy, where the FeCl₃ initiator was converted to [FeCl₄] unit having OER activity in PTh, PANI and PPy matrix. These conductive polymers possess great hole extraction and storage capabilities, enhancing charge carrier separation and accelerating OER kinetics due to the presence of [FeCl₄] units, which presented as functions of supramolecular catalyst. Among them, the PTh/BVO reaches the highest photocurrent density of 4.72 mA cm⁻² at 1.23 V vs. RHE under AM 1.5 G illumination with a better ABPE of 1.34% at 0.75 V vs. RHE. Moreover, the PTh layer with good anti-corrosion of seawater and suppression of Cl⁻ oxidation by [FeCl₄] units makes BVO great stability for PEC seawater splitting. We believe that these new findings can provide a proof-of-concept for ligand design engineering in OECs.

## Methods

### Preparation of BiVO₄ (BVO) photoanodes

BiVO₄ photoanodes were fabricated by referring to Choi's method[19]. In detail, 50 mL of the precursor solution containing 0.4 M KI and 0.04 M Bi (NO₃)₃•5H₂O was adjusted pH to 1.6 - 1.8 by adding HNO₃. This solution was mixed with 20 mL of ethanol containing 0.23 M p-benzoquinone and was vigorously stirred for a few minutes. A typical three-electrode cell was used for electrodeposition where a piece of F-doped SnO₂ coated glass (FTO, 1×2 cm) served as the working electrode, a Pt electrode served as the counter electrode and an Ag/AgCl electrode served as the reference electrode. The cathodic deposition was conducted potentiostatically at −0.1 V vs. Ag/AgCl for 300 s at room temperature. Then, 0.2 mL DMSO solution containing 0.2 M VO(acac)₂ was placed on the reddish brown BiOI electrode, and annealed in air at 450 °C for 2 h with ramping rate of 2 °C min⁻¹. The BiVO₄ electrode was immersed in 1 M NaOH solution for 30 minutes to remove excess V₂O₅. The resulting bare BiVO₄ electrodes were rinsed with deionized water and dried in air.

### Preparation of PTh/BVO photoanodes

The BVO photoelectrode rinsed with deionized water was immersed in 50 ml acetonitrile solution containing 50 mM thiophene monomer for 2 h. Subsequently, 50 ml of FeCl₃ solution (0.12 M) was added and in-situ polymerized for 24 h. After that, the PTh/ BVO electrode was removed, rinsed with methanol, and dried at room temperature.

### Preparation of PPy/BVO and PANI/BVO photoanodes

The PPy/BVO and PANI/BVO photoanode were prepared in a similar way as described above, except that the monomers were replaced with pyrrole and aniline, chemically polymerized for different times, and subsequently dried at room temperature.

### Preparation of PTh/TiO₂ and PTh/WO₃ photoanodes

The TiO₂ photoanodes were synthesized using a hydrothermal method according to a previous report[63]. In brief, 0.5 mL titanium butoxide was added to a 50 mL solution containing deionized water and concentrated HCl (37%) with a 1:1 ratio. After stirring for a few minutes, the FTO substrates were placed in a Teflon liner and heated to 180 °C for 12 h. The obtained TiO₂ photoanodes were washed and annealed at 400 °C for 2 h in air. The PTh/TiO₂ preparation is similar to PTh/BVO.

### Preparation of PTh/TiO₂ and PTh/WO₃ photoanodes

The WO₃ photoanodes were prepared according to a previously reported method[54]. Firstly, the WO₃ seed layer was prepared by spin-coating and annealing on the FTO substrate. Second, the seed layer was hydrothermally heated at 180 °C for 8 h in a mixture containing Na₂WO₄ • 2H₂O, concentrated HCl, oxalic acid, and DI water. Then, after washing and drying, the samples were annealed at 500 °C for 2 h. The PTh/WO₃ Preparation is similar to PTh/BVO.

### Photoelectrochemical measurements

The photoelectrochemical (PEC) performances of all photoanodes were evaluated in a standard three-electrode configuration cell with a platinum wire as the counter electrode, an Ag/AgCl reference electrode as a reference electrode, and 0.5 M potassium borate (pH 9.0) used as the electrolyte for all PEC measurements. Data were collected on a CHI 660E electrochemical workstation. The illumination source was a 300 W Xe arc lamp with AM 1.5 G filter (100 mW/cm², FX 300, Beijing PerfectLight Co. Ltd). Linear sweep voltammetry (LSV) was performed by sweeping the potential in the positive direction at a scan rate of 20 mV/s. All the potentials versus reversible hydrogen potential (vs. RHE) were converted from the potentials versus Ag/AgCl

according to the Nernst equation:

$$E_{RHE} = E_{Ag/AgCl} + 0.059pH + 0.197 \quad (3)$$

The ABPE can be calculated by the photocurrent-potential curves under chopped AM 1.5 G sunlight using the equation:

$$ABPE(\%) = [(1.23 - V_{app}) \times (J_{light} - J_{dark})/P_{light}] \times 100\% \quad (4)$$

where $V_{app}$ is the applied potential, $J_{dark}$ and $J_{light}$ are the photocurrent density (mA cm$^{-2}$) in darkness and under AM 1.5 G light, and $P_{light}$ is the power density of AM 1.5 G (100 mW cm$^{-2}$).

The IPCE was obtained using a monochromator coupled with a 300 W Xe lamp as the simulated light source. An applied potential of 1.23 V vs. RHE was supplied by a CHI660e electrochemical workstation and the power density at a specific wavelength was measured by a Newport power meter. IPCE values were calculated using the following equation:

$$IPCE(\%) = [1240 \times J_{light}/\lambda \times P_{light}] \times 100\% \quad (5)$$

where $J_{light}$ is the photocurrent density (mA cm$^{-2}$) obtained from the electrochemical workstation, $\lambda$ is the wavelength (nm) of incident radiation, and $P_{light}$ is the power density obtained at a specific wavelength (mW cm$^{-2}$). The wavelength dependence of IPCE was measured under monochromatic irradiation from a 300 W Xe lamp equipped with bandpass filters.

Electrochemical impedance spectroscopy (EIS) measurements were carried out with a sinusoidal AC perturbation of 5 mV applied over the frequency range of $10^{-2}$ Hz-$10^6$ Hz. Mott–Schottky (MS) curves were measured with a frequency of 500 Hz and an amplitude of 10 mV. From the slope and intercept of the Mott-Schottky plots, the charge carrier densities (N$_D$) were estimated using the following equation:

$$\frac{A^2}{C^2} = \frac{2}{\varepsilon_r\varepsilon_0 qN_D}V - \frac{2\left(E_{fb} + k_BT/q\right)}{\varepsilon_r\varepsilon_0 qN_D} \quad (6)$$

where $C$ is the capacitance, $A$ is the effective surface area, $V$ is the voltage applied to the electrode, $E_{fb}$ is the flat band potential, $q$ is the charge of an electron ($1.60 \times 10^{-19}$ C), $\varepsilon_O$ is the permittivity of vacuum ($8.854 \times 10^{-14}$ F cm$^{-1}$), $\varepsilon_r$ is the dielectric constant of semiconductor (68 for BiVO$_4$).

Intensity-modulated photocurrent spectroscopy (IMPS) was measured by Zahner electrochemical workstation, the intensity of the light source used is 30 W m$^{-2}$. The hydrogen and oxygen evolution were detected by a gas chromatograph (PerkinElmer, Clarus 580, USA, TCD, 5 Å molecular sieve columns and Ar carrier) every 2 h.

## Materials characterization

The $^{57}$Fe Mössbauer spectra of polythiophene were recorded on a SEE Co W304 Mössbauer spectrometer, using a $^{57}$Co/Rh source in transmission geometry. The data were fitted by using the MossWinn 4.0 software. TOF-SIMS was performed on the TOF-SIMS 5-100 (ION-TOF GmbH) equipped with a reflection-type TOF analyzer. A pulsed Bi primary ion beam source at an acceleration voltage of 30 keV, 100 μm×100 μm raster size. SEM images were taken on a FEI Quanta 250FEG scanning electron microscope. HR-TEM images were collected on a JEOL JEM-AFM 200 F (Japan) electron microscope with (Cs-corrected/energy-dispersive X-ray spectroscopy (EDS)/EELS). X-ray diffraction (XRD) patterns were acquired on a Bruker-AXS D8 Advance diffraction system equipped with a CuKα source (λ = 1.54056 Å). UV-vis absorbance spectra were collected on a Shimadzu UV-3600 spectrometer equipped with an integrating sphere. XPS was recorded on a XPS instrument (Thermo Scientific K-Alpha + ) equipped with an aluminium anode (Al Kα, λ = 1486.6 eV), and all the peaks were calibrated with a C

1 s spectrum at a binding energy of 284.8 eV. PL and TR-PL were taken on Edinburgh FLS1000 equipped with an Oxford cryostat designed for optical measurement. The transient-state surface photovoltage (TS-SPV) measurements were carried out on a home-made system. The samples were excited with a laser radiation pulse (wavelength of 355 nm and pulse width of 5 ns) from a third-harmonic Nd:YAG laser (Polaris II, New Wave Research, Inc.). The TS-SPV signal was recorded by using a 500 MHz digital phosphor oscilloscope (TDS 5054, Tektronix). The effective surface areas of the as-prepared samples were measured by nitrogen adsorption-desorption isotherms on the Micrometrics ASAP 2020 system, employing a fitting analysis based on the Brunauer-Emmett-Teller (BET) equation.

## Computational details

All calculations were performed with density functional theory (DFT) methods implemented in the Vienna ab initio simulation package (VASP) for the exchange-correlation functional of generalized gradient approximation and the energy cutoff is set to 300 eV for the plane wave. The convergence criterion for electronic relaxation is set to be $1 \times 10^{-4}$ eV and the geometries are optimized until the force on each ion less than 0.1 eV/Å. The zero-point energy and entropy effect on Gibbs free energies was derived from frequency calculation and treated with VASPKIT code.

The adsorption energy of $E_{ads}$ was calculated based on the following formulas within the ox-red potential:

$$\begin{aligned}E_{ads}(Cl^-) &= E(^*Cl^-) - E(^*) - 1/2E(Cl_2 - 1.396).E_{ads}(OH^-)\\ &= E(^*OH) - E(^*) - [E(H_2O) + 0.828 - 1/2E(H_2)]\end{aligned} \quad (7)$$

In our model, we assumed that the OER is processed in the four electrons pathway,

$$H_2O(l) + {}^* \rightarrow OH^* + e^- + H^+ \quad (8)$$

$$OH^* \rightarrow O^* + e^- + H^+ \quad (9)$$

$$H_2O(l) + O^* \rightarrow OOH^* + e^- + H^+ \quad (10)$$

$$OOH^* \rightarrow O_2(g) + e^- + H^+ \quad (11)$$

Where * referred to an active site on the catalyst, (l) and (g) represented liquid and gas phases, respectively. O*, OH* and OOH* were adsorbed intermediates. For each step, the reaction free energy (ΔG) was defined as the difference between the free energies of the initial and final states.

## Data availability

The data described in this paper are available from the authors upon reasonable request.

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

## Acknowledgements
This work was supported by the NSFC (T2322013, 22025202), the Natural Science Foundation of Jiangsu Province of China (BK 20220079). The authors thank Dr. Xiangyu Guo, who participated in the discussions for analyzing the data obtained from the calculations.

## Author contributions
Z.S.L. and K.Z. conceived and designed the experiments. J.M.M. carried out materials synthesis and electrochemical characterization. Y.Y., Y. A. and C.L. participated in part of the synthesis. X. J. Y. carried out the theoretical calculation. K.Z., J.M.M. and Z.S.L. co-wrote and supervised the paper. All authors discussed the results and commented on the manuscript.

## Competing interests
The authors declare no competing interests.
