## [Peer review file · Nature Communications]

Supramolecular Catalyst with [FeCl₄] Unit Boosting Photoelectrochemical Seawater Splitting via Water Nucleophilic Attack PathwayREVIEWER COMMENTS

Reviewer #1 (Remarks to the Author):

In the manuscript, the authors proposed a strategy to improve charge transfer at interfaces of photoanode using supramolecular polymers. Although the photocurrent density is not the highest, the strategy is novel and interesting and may be able to address the challenge of hybridizing interfaces of molecular catalysts and solar absorbers. Moreover, the characterizations are enough to support their main conclusions, therefore I would like suggest publication after minor revisions.

-The authors used transient surface photovoltage (TPV) measurements to indicate the charge separation and transfer process, yet the explanation is not very clear, especially in the (2) and (3) stages. The author should make the necessary additional remarks.

-The authors used a PTh/BVO photoanode to achieve PEC seawater splitting with high stability, and put forward a dynamic recycling mechanism of Cl ions. However, this process involving chemical absorption requires further evidence to confirm the mechanism, such as XPS spectra of Cl on PTh/BVO photoanode.

-The authors used polythiophene as a model to inhibit seawater corrosion, while, do other similar structures have the same effect? (Such as PPy and PANI mentioned by the authors)

-The author's description of the polymerization process of thiophene monomer after the introduction of FeCl₃ is slightly less, and it is suggested to increase the discussion appropriately.

-Please unify the abbreviations of BiVO₄ or BVO.

--Based on my experience, the pristine BVO has a relatively well stability in seawater. It can be quickly decayed at the initial and then keep a relatively stable photoreponse for a certain time. The result of BVO shown in Fig. 4b may be further extended the measured time to observe.

-Page 4, Line 71, sentence of "naturally formed the polymerization process" lacks preposition, Page 17, Line 16: The lapsus calami "5h" should be corrected to "40h".

-The pH of the seawater used in the author's experiments may have been biased and a correction is recommended.

Reviewer #2 (Remarks to the Author):

This work reports the synthesis of porous BiVO₄ photoanode with surface coupled by [FeCl₄] water oxidation active sites encapsulated in polythiophene matrix. A photocurrent of 4.72 mA cm⁻² was

achieved at 1.23 V vs. RHE using simulated solar light (AM1.5G). Excellent robustness was realized on this photoanode for photoelectrochemical (PEC) seawater splitting because of the suppressed Cl₂ generation by [FeCl₄], the anti-corrosion capacity by polythiophene together with the intimately coupled interface between BiVO₄ and [FeCl₄] enclosed in polythiophene matrix. This work is creative and shows some significance on PEC splitting of natural seawater. So I recommend acceptance after resolving the issues below:

1. Is this strategy (coated with [FeCl₄] enclosed in polythiophene matrix) a general one which can be used for the other photoanodes, such as TiO₂ and WO₃? Authors should show more results on the universality of this strategy to highlight the significance of this work.
2. Authors claim that [FeCl₄] serves as the active sites for O₂ evolution and suppress Cl₂ evolution. Can Authors show some insights into this mechanism using theoretical computations? Authors may refer some recent publications about seawater splitting such as Science Advances 2023, 9, eadi7755. Angew. Chem. Int. Ed. 2023, 62, DOI: 10.1002/anie.202311674. Nature Energy 2021, 6, 904-912.
3. In Supplementary Fig. 5 and Fig. 31, there is still one peak near Fe 2p_{3/2} not identified. Authors should explain it.
4. Results on the kinetics of photo-excited electrons and holes in/on the surface of this photoanode should be added to confirm that this interface engineering strategy could efficiently boost the extraction of photo-excited holes, especially when using the seawater.
5. What about the results on the scale-up of this strategy for realistic application? Can Authors add results on this issue?
6. How about the stability of pth after the stability test? Is there any change on the structure of pth after the stability test? Authors should show results on the structure stability of pth.

Reviewer #3 (Remarks to the Author):

In this study Miao et al investigate the activity, and foremost the charge transport properties, of BVO photoelectrodes modified with organic/inorganic co-catalyst layers consisting of polymerized polythiophene with FeCl₄ catalytic centers, towards water splitting.

Especially the thorough and advanced photoelectrochemical characterization of the system is noteworthy, providing a complete and detailed picture of the charge transport properties of the PTh layer, the Fe-based centers, and how these compare to pristine materials. This convincingly shows that the organic/inorganic Pth-FeCl₄ hybrid layer acts as a co-catalyst that can improve the OER by facilitating the charge transfer process.

I am less impressed, however, by the novelty of this work. While it is claimed this is the first of its kind (line 25), it is not fully clear how this supramolecular structure is so different from earlier reports on BVO electrodes, that were too modified with a polymer charge extraction layer and Fe-based centers, which is topically extremely close to the present work (see for instance J. Mater. Chem. A, 2020, 8, 16283-16290).

In addition, what is really lacking from the present dataset, is experimental proof of the actual water splitting activity. Only photoelectrochemical tests have been performed, but what are the actual H₂ and O₂ yields, at which Faradaic efficiency, and how do these performances compare to existing materials?

Especially these above two points should be addressed in depth before further considering this manuscript.

Furthermore, there are still several minor remarks that can be considered when reworking the manuscript for resubmission (elsewhere):

- The manuscript is not well written. While this might not be seen as a scientific comment, I must point out that at some instances the meaning of certain sentences was simply not captured due to this style of writing. For instance:

line 54-56: “photo-induced charges always need across the interface formed by a molecular anchoring that is sensitive to the surrounding chemical environment”??

or line 269-270: “corroborating that [FeCl₄] unit as electron donor robust the electronic conductivity”??

- Line 93: If Fe is in the trivalent form, then FeCl₄ should be an anion. However, it is nowhere denoted as [FeCl₄]⁻¹

- Line 140: pristine “BVO exhibits a three-dimensional 2nanoporous...” Is the ‘2’ a typo? Also, ‘nanoporous’ is not IUPAC terminology; a material is either microporous, mesoporous or microporous.

- I believe the results listed in Table S1 are important enough to be moved to the main text.

- Line 285-287: Different reaction orders are found for PTh/BVO and BVO. Based on this it is concluded that PTh/BVO exhibits faster water oxidation kinetics. However, if different materials exhibit different reaction orders for a given reaction, their rates cannot be readily compared due to the fact that the rate constants have different units. Hence, this statement is scientifically not sound.

- The authors performed experiments with ‘natural seawater’. In that case it is quite important to mention the source, and to report the mineral composition of this seawater. Especially the presence of divalent cations in seawater can have a detrimental impact on the performance, and these can differ from site to site.

Response to the Comments on Nature Communications

Manuscript ID: NCOMMS-23-48472

Dear Reviewers:

We would like to thank the reviewers for careful reading and helpful comments. We revised the manuscript thoroughly according to the comments. The added items are highlighted in red in the main manuscript and the supplementary information. The following changes were made and listed below:

Reviewer #1

“In the manuscript, the authors proposed a strategy to improve charge transfer at interfaces of photoanode using supramolecular polymers. Although the photocurrent density is not the highest, the strategy is novel and interesting and may be able to address the challenge of hybridizing interfaces of molecular catalysts and solar absorbers. Moreover, the characterizations are enough to support their main conclusions, therefore I would like suggest publication after minor revisions.”

Response: Thank you very much for your positive evaluation of our work. We believe that your valuable suggestions will improve the quantity of this manuscript.

Comments 1:

“The authors used transient surface photovoltage (TPV) measurements to indicate the charge separation and transfer process, yet the explanation is not very clear, especially in the (2) and (3) stages. The author should make the necessary additional remarks.”

Response: Thank you for the detailed review. The TPV measurement is an intuitive approach to represent the kinetics of photo-generated charge carriers, especially it can monitor the process of charge separation and transfer. Considering the unique chemical structure at the hybridization interface, we have added some descriptions as follows:

For the (2) stage, a minority of the surface holes are trapped by the surface states, so the photovoltage signal has a small period of sharp drop (1.2×10^{-7} to 4.0×10^{-7} s); For the (3) stage, the Fe sites located in the PTh matrix stores the injected holes by their changeable valence, and built-in electric field mitigates the recombination of charge carriers, so that these holes have long lifetime and avoid fast charge recombination, which is displayed as a flattening of the photovoltage signal and reaching a stable intensity (2.0×10^{-6} s, Peak 2)

Comments 2:

“The authors used a PTh/BVO photoanode to achieve PEC seawater splitting with high stability, and put forward a dynamic recycling mechanism of Cl ions. However, this process involving chemical absorption requires further evidence to confirm the mechanism, such as XPS spectra of Cl on PTh/BVO photoanode.”

Response: Thanks for your great suggestion. We added XPS spectra to further analyze the dynamic recycling mechanism of Cl ions, and the relevant description has been added in the revised manuscript. It can be seen that there are slight changes in the process, which will make the process easier to understand.

To determine the Cl⁻ storage properties of polythiophene, the Cl 2p XPS spectra after long-term stability testing were investigated. As shown in Supplementary Fig. 37, the peak at 200.1 eV associated with the stored Cl ions is ascribed to the charge-transfer interaction between the Cl ions and the polythiophene chain. It can be seen that such a peak disappears when the i-t testing was performed in the KBi solution, while the peak intensity slightly increases after the i-t testing in seawater.

Fig. R1. Cl 2p XPS spectra of PTh/BVO before and after i-t testing in KBi and seawater. (This Figure is included as Supplementary Fig. 37 in the revised manuscript)

Comments 3:

“The authors used polythiophene as a model to inhibit seawater corrosion, while, do other similar structures have the same effect? (Such as PPy and PANI mentioned by the authors).”

Response: Thanks for your nice suggestion. We prepared PPy/BVO and PANI/BVO photoanodes, carried out stability tests in a seawater environment, and plotted the results. As shown in Fig. R2, both electrodes exhibit good PEC seawater splitting stability, confirming that our proposed strategy is universal and effective.

Fig. R2. Long-term PEC stability of PPy/BVO and PANI/BVO at 1.23 V vs. RHE in seawater electrolyte (pH = 8.1) under AM 1.5G illumination. (This Figure is included as Supplementary Fig. 42 in the revised manuscript)

Comments 4:

“The author's description of the polymerization process of thiophene monomer after the introduction of FeCl₃ is slightly less, and it is suggested to increase the discussion appropriately.”

Response: Thank you for your valuable advice. We have added the discussion as follows:

Briefly, FeCl₃ acts as an oxidant, which can oxidize the thiophene monomer to form cationic radicals. Then, these cationic radicals are polymerized into long chains for a spontaneous polymerization process.

Comments 5:

“Please unify the abbreviations of BiVO₄ or BVO.”

Response: Thank you for the detailed review. We unified the words in the revised manuscript.

Comments 6:

“Based on my experience, the pristine BVO has a relatively well stability in seawater. It can be quickly decayed at the initial and then keep a relatively stable photoreponse for a certain time. The result of BVO shown in Fig. 4b may be further extended the measured time to observe.”

Response: Thank you for your feedback. We have added the results as follows and made the change in the manuscript (Fig. R3)

Fig. R3. Long-term PEC stability of PTh/BVO and BVO at 1.23 V vs. RHE in seawater electrolyte (pH = 8.1) under AM 1.5G illumination. (This Figure is included as Fig. 4a in the revised manuscript)

Comments 7:

“Page 4, Line 71, sentence of “naturally formed the polymerization process” lacks preposition, Page 17, Line 16: The lapsus calami “5h” should be corrected to “40h”.”

Response: Thank you for the detailed review. We have corrected these mistakes.

Comments 8:

“The pH of the seawater used in the author's experiments may have been biased and a correction is recommended.”

Response: Thank you for your detailed feedback. We have re-determined the pH and main components of the seawater (Table R1) and placed them in Supplementary Table 2.

Table R1. The pH and main components of the seawater in this work.

Seawater	
(from Tsingtao, China; pH=8.1)	
Cations	Concentration (mg/L)
Na	9731
Mg	993
K	297
Ca	332
Sr	4.48

Reviewer #2

“This work reports the synthesis of porous BiVO₄ photoanode with surface coupled by [FeCl₄] water oxidation active sites encapsulated in polythiophene matrix. A photocurrent of 4.72 mA cm⁻² was achieved at 1.23 V vs. RHE using simulated solar light (AM1.5G). Excellent robustness was realized on this photoanode for photoelectrochemical (PEC) seawater splitting because of the suppressed Cl₂ generation by [FeCl₄], the anti-corrosion capacity by polythiophene together with the intimately coupled interface between BiVO₄ and [FeCl₄] enclosed in polythiophene matrix. This work is creative and shows some significance on PEC splitting of natural seawater. So I recommend acceptance after resolving the issues below:”

Response: Thank you for your encouragement and positive comments. According to your comments, we have revised our manuscript to address all your concerns.

Comments 1:

“Is this strategy (coated with [FeCl₄] enclosed in polythiophene matrix) a general one which can be used for the other photoanodes, such as TiO₂ and WO₃? Authors should show more results on the universality of this strategy to highlight the significance of this work.”

Response: Thank you for your valuable suggestions. The supramolecular polymerization strategy has been applied to the modification of TiO₂ and WO₃ electrodes, which show similar effect on PEC performance enhancement (Fig. R4-5).

Fig. R4. LSV curves of PTh/TiO₂ and TiO₂ photoanode in KBi (pH=9) under (a) AM1.5G and (b) dark. SEM images of (c) TiO₂ and (d) PTh/TiO₂. (e) Corresponding EDS element mapping images. (This Figure is included as Supplementary Fig. 43 in the revised manuscript)

Fig. R5. LSV curves of PTh/WO₃ and WO₃ photoanode in KBi (pH=9) under (a) AM1.5G and (b) dark. SEM images of (c) WO₃ and (d) PTh/WO₃. (e) Corresponding EDS element mapping images. (This Figure is included as Supplementary Fig. 44 in the revised manuscript)

Comments 2:

“Authors claim that [FeCl₄] serves as the active sites for O₂ evolution and suppress Cl₂ evolution. Can Authors show some insights into this mechanism using theoretical computations? Authors may refer some recent publications about seawater splitting such as *Science Advances* 2023, 9, eadi7755. *Angew. Chem. Int. Ed.* 2023, 62, DOI: 10.1002/anie.202311674. *Nature Energy* 2021, 6, 904-912.”

Response: Thanks for your great suggestion on improving the accessibility of our manuscript. We have supplemented the theoretical calculations and added relevant discussion as follows:

As shown in Fig. R6, the adsorption energies of Cl⁻ and OH⁻ on [FeCl₄] unit are theoretically calculated. The adsorption energy of Cl⁻ is 0.690 eV, indicating that the process is endothermic and does not easily occur, while the adsorption energy of OH⁻ is -0.063 eV, which was more inclined to be adsorbed spontaneously.

Fig. R6. Atomic structures of Cl^- and OH^- adsorption on $[\text{FeCl}_4]$ unit. (This Figure is included as Fig. 4b in the revised manuscript)

Due to the easier adsorption for OH^- , the WNA mechanism occurs more tendentiously during the water oxidation, during which OH^- continuously nucleophilic attacks to complete the proton-electron coupling process via density functional theory (DFT) calculations (Fig. R7). For the BVO sample, the second step ($^*\text{OH} \rightarrow ^*\text{O}$) is the rate-determining step with about 0.99 eV overpotential, leading to the sluggish OER kinetics. For PTh/BVO, both the OH^* and O^* steps exhibit low Gibbs free energies, and the lower overpotential with 0.69 eV of the rate-determining step ($\text{O}^* \rightarrow \text{OOH}^*$) indicates the optimized water oxidation kinetics.

Fig. R7. Schematics and free energy diagram for OER pathway on PTh/BVO. (This Figure is included as Fig. 4c in the revised manuscript)

Comments 3:

“In Supplementary Fig. 5 and Fig. 31, there is still one peak near Fe 2p_{3/2} not identified. Authors should explain it.”

Response: Thank you for your comments. We checked the original data, and we considered that the deconvoluted peak appearance should be caused by the uncompleted removal of divalent iron which contaminated samples. So, we retested the XPS by preparing samples as those measured for the Mossbauer spectra. The XPS spectra were re-fitted to present trivalent iron only (Fig. R8).

Fig. R8. Fe 2p XPS spectra of (a) PTh and PTh/BVO and (b) PTh/BVO before and after stability testing in seawater. (This Figure is included as Supplementary Fig. 5 and Fig. 35 in the revised manuscript)

Comments 4:

“Results on the kinetics of photo-excited electrons and holes in/on the surface of this photoanode should be added to confirm that this interface engineering strategy could efficiently boost the extraction of photo-excited holes, especially when using the seawater.”

Response: Thank you for your valuable advice. We have introduced the kinetics of carrier's separation and transfer in detail using TPV, IMPS measurements, etc.. However, as you mentioned, they may be different in seawater, hence we have measured the transfer efficiency (Also known as charge injection efficiency) in seawater to help better understand the kinetics of photo-excited charge (Fig. R9). As expected, the transfer efficiency of photo-excited charge is greatly enhanced, demonstrating the accelerated carrier kinetics.

Fig. R9. Charge transfer efficiency of PTh/BVO and BVO in PEC seawater splitting. (This Figure is included as Supplementary Fig. 28 in the revised manuscript)

Comments 5:

“What about the results on the scale-up of this strategy for realistic application? Can Authors add results on this issue?”

Response: Thanks for your kind advice. According to your suggestion, we attempted to prepare a BVO photoanode as large as possible, and finally, we achieved a uniform BVO photoanode with an area of about 30 cm². Then, we adopted the in-situ polymerization strategy to construct the PTh/BVO photoelectrode. After the PEC test of the photoanode, it is found that it can achieve high photocurrent and stability (Fig. R10), and it has a similar enhancement factor to the small-sized one.

Fig. R10. (a) Digital photographs of PTh/BVO photoanodes with different areas. (b) PEC performance of PTh/BVO and BVO with 30 cm². (c) Stability testing of PTh/BVO at 1.23 V vs. RHE in seawater electrolyte (pH = 8.1) under AM 1.5G illumination. (This Figure is included as Supplementary Fig. 38 in the revised manuscript)

Comments 6:

“How about the stability of pth after the stability test? Is there any change on the structure of pth after the stability test? Authors should show results on the structure stability of pth.”

Response: Thanks for your kind advice. In our original manuscript, we provided a series of characterizations to demonstrate the structure stability of PTh/BVO photoanode, as shown in Supplementary Fig. 32. To further determine the structural stability of PTh, the FTIR spectra of PTh/BVO before and after the stability test were measured, which further confirm unchanged polymer structure (Fig. R11).

Fig. R11. (a) TEM images of PTh/BVO after stability test. (b) FTIR spectra of PTh/BVO before and after stability testing. (This Figure is included as Supplementary Fig. 32 and Fig. 33 in the revised manuscript)

Reviewer #3

“In this study Miao et al investigate the activity, and foremost the charge transport properties, of BVO photoelectrodes modified with organic/inorganic co-catalyst layers consisting of polymerized polythiophene with FeCl₄ catalytic centers, towards water splitting. Especially the thorough and advanced photoelectrochemical characterization of the system is noteworthy, providing a complete and detailed picture of the charge transport properties of the PTh layer, the Fe-based centers, and how these compare to pristine materials. This convincingly shows that the organic/inorganic PtH-FeCl₄ hybrid layer acts as a co-catalyst that can improve the OER by facilitating the charge transfer process.”

Response: Thank you for your encouragement and positive comments, which may make our paper more instructive. We have taken your comments seriously and made changes after thinking deeply.

Comments 1:

“I am less impressed, however, by the novelty of this work. While it is claimed this is the first of its kind (line 25), it is not fully clear how this supramolecular structure is so different from earlier reports on BVO electrodes, that were too modified with a polymer charge extraction layer and Fe-based centers, which is topically extremely close to the present work (see for instance J. Mater. Chem. A, 2020, 8, 16283-16290).”

Response: Thank you for your comments on the novelty of our work, which are valuable and very helpful for revising and improving our paper. We have studied the nice reference mentioned carefully, but we think that it is totally different from our case. In that paper, the role of polymer is light absorber which can form a Z-type heterojunction to improve the charge separation efficiency of BVO. The organic semiconductor hybridized with inorganic semiconductor to construct a heterojunction has been well-documented so far. For PEC related fields, there are some reports on the polymer's strategies to modify photoanodes, in addition to enhancing the charge separation efficiency, some of which gave play to photothermal effects (*J. Mater. Chem. A*, 2020, 8, 15976), passivation (*J. Colloid Interface Sci*, 2023, 644, 533-545) and so on.

In our work, we presented a different role of polymers from all other reports. We introduce a stable [FeCl₄] unit in the polymer matrix by a simple initiator effect, which is universal. The [FeCl₄] unit in the polymer matrix made it have great catalytic activity of oxygen evolution reaction. We named it a supramolecular-like catalyst, as it not only

is a molecule structure but also is a single active site. More importantly, the $[\text{FeCl}_4]$ unit can effectively suppress Cl ion adsorption, enabling stable performance for PEC seawater splitting. This work may provide a potential solution for solving the problem of molecular catalyst/photo-absorber hybridization interface.

Comments 2:

“In addition, what is really lacking from the present dataset, is experimental proof of the actual water splitting activity. Only photoelectrochemical tests have been performed, but what are the actual H₂ and O₂ yields, at which Faradaic efficiency, and how do these performances compare to existing materials?”

Especially these above two points should be addressed in depth before further considering this manuscript.”

Response: Thanks for your great suggestion. We carried out a quantitative gas test (Fig. R12), and the amount of oxygen and hydrogen produced after 10 hours was approximately 795.8 μmol and 369.2 μmol , respectively, corresponding to a Faraday efficiency of up to 91.7 %. The results are comparable with other similar work on PEC seawater splitting (Table R2). All results are added in the Supplementary Information.

Fig. R12. Time course of O₂ and H₂ gas evolution of PEC seawater splitting using PTh/BVO photoanode. (This Figure is included as Supplementary Fig. 29 in the revised manuscript)

Table R2. Comparison of the Faradic efficiency (H₂) in seawater electrolyte with different photoanode materials. (As Supplementary Table 3 in the revised manuscript)

Photoanode	Faradic efficiency (H ₂)	Ref.
Ti-Fe ₂ O ₃ -In ₂ O ₃ /CoOOH	90%	1
Ag/WO ₃ /ZnFe-LDH	near 100%	2
SrNbO ₂ N/Nb	90%	3
WO ₃	80%	4
Co:BiVO ₄	92%	5
PTh/BVO	91.7%	This work

References in Table R2:

1 ACS Appl. Nano Mater. 2023, 6, 20240-20250.

2 Appl. Catal., B 2020, 264, 118540.

3 ACS Sustainable Chem. Eng. 2023, 11, 1655-1665.

4 Adv. Energy Mater. 2020, 10, 1903213.

5 Dalton Trans. 2023, 52, 2051-2061.

Comments 3:

“Furthermore, there are still several minor remarks that can be considered when reworking the manuscript for resubmission (elsewhere):

- The manuscript is not well written. While this might not be seen as a scientific comment, I must point out that at some instances the meaning of certain sentences was simply not captured due to this style of writing. For instance:

line 54-56: “photo-induced charges always need across the interface formed by a molecular anchoring that is sensitive to the surrounding chemical environment”??

or line 269-270: “corroborating that [FeCl₄] unit as electron donor robust the electronic conductivity”??”

Response: Thanks for your kind advice. We have corrected these sentences as follows to facilitate the reader's understanding.

For the first question: “....., as the photo-induced charges always need across the interface formed by a molecular anchoring. **The anchor groups are always sensitive to the surrounding chemical environment, and therefore constraining the interfacial stability.**”

For the second: “which was 5.8 times higher than that of BVO ($6.87 \times 10^{18} \text{ cm}^{-3}$), corroborating that $[\text{FeCl}_4]$ unit robust the charge transport process.”

Comments 4:

“Line 93: If Fe is in the trivalent form, then FeCl_4 should be an anion. However, it is nowhere denoted as $[\text{FeCl}_4]^{-1}$ ”

Response: Thanks for your nice comment. We have mentioned its ionic properties in the previous section, which are presented in the first part of the manuscript. Since this ion will form weak coordination with S in the PTh matrix and we regard it as a catalytic unit in this paper, thus it is denoted as $[\text{FeCl}_4]$ for more convenience and intuition.

Comments 5:

“Line 140: pristine “BVO exhibits a three-dimensional 2nanoporous...” Is the ‘2’ a typo? Also, ‘nanoporous’ is not IUPAC terminology; a material is either microporous, mesoporous or microporous.”

Response: Thanks for your nice comment. We are very sorry for this typo, and we have made corrections based on your comments. It has been reported in the past that the BVO in this paper contains microporous and mesoporous (*Science*, 2014, 343, 990-994; *Appl. Catal. B*, 2018, **225**, 504-511), so we revised this inaccurate word to “porous”.

Comments 6:

“I believe the results listed in Table S1 are important enough to be moved to the main text.”

Response: Thanks for your great suggestion, which will highlight the importance of the TRPL measurement. We have moved it as Table 2 to the main text.

Comments 7:

“Line 285-287: Different reaction orders are found for PTh/BVO and BVO. Based on this it is concluded that PTh/BVO exhibits faster water oxidation kinetics. However, if different materials exhibit different reaction orders for a given reaction, their rates cannot be readily compared due to the fact that the rate constants have different units. Hence, this statement is scientifically not sound.”

Response: Thanks for the nice comments. We have thought carefully about your suggestions, and indeed, as you mentioned, although the reaction order can reflect the

pathway of water oxidation, it cannot be compared in terms of kinetics. We have made corrections to the manuscript. Our demonstration can only explain the difference in the hole transfer pathway of the PTh/BVO and BVO photoanode. By calculating the reaction order, the results show a WNA mechanism for PTh/BVO, which is consistent with the results expected from the single catalytic site, and thus we claim it can be considered a kind of molecular catalysis mechanism. In addition, to compare the different kinetics of water oxidation, we performed KIE measurements at later passages of the manuscript, which would determine the reaction rates.

Comments 8:

“The authors performed experiments with ‘natural seawater’. In that case it is quite important to mention the source, and to report the mineral composition of this seawater. Especially the presence of divalent cations in seawater can have a detrimental impact on the performance, and these can differ from site to site.”

Response: Thanks for your kind advice. The seawater selected for our work came from Tsingtao City, China, and was filtered for the experiment. The main cationic components of seawater electrolytes were measured by ICP-MS and are listed in Table R3. Although Ca^{2+} , Mg^{2+} and other cations may have an impact on the PEC system, while, Cl^- is the most abundant ion in seawater. Our work is carried out based on the inhibition of Cl^- , thus demonstrating its universality.

Table R3. The pH and main components of the seawater in this work. (As Supplementary Table 2 in the revised manuscript)

Seawater	
(from Tsingtao, China; pH=8.1)	
Cations	Concentration (mg/L)
Na	9731
Mg	993
K	297
Ca	332
Sr	4.48

REVIEWERS' COMMENTS

Reviewer #1 (Remarks to the Author):

The authors have answered my questions.

Reviewer #2 (Remarks to the Author):

not comments

Reviewer #3 (Remarks to the Author):

The authors have duly considered the comments raised by the various reviewers.

In addressing my comment on the amount of H₂ and O₂ being generated, I believe the authors accidentally switched the values of H₂ and O₂ in the main text on page 17: "To understand the CER inhibition via the PTh/BVO, the amounts of O₂ and H₂ products during PEC seawater splitting of 10 hours were measured, which are approximately 795.8 and 369.2 μmol , respectively, (Supplementary Fig. 29)." => There should be more H₂ than O₂, so these values need to switch places.

Finally, I would still strongly advise the authors to consult a professional language editor. The level of English employed remains poor. In my review I gave just two examples, representative for the rest of the text. Instead of scrutinizing the entire manuscript as suggested, the authors have addressed these two specific lines, and still even these sentences remain intelligible. For instance: "...photo-induced charges always need across the interface formed by molecular anchoring"?

Hence, I recommend acceptance of this manuscript from a scientific perspective, but quite some editing will still be required.

Response to the Comments on Nature Communications

Manuscript ID: NCOMMS-23-48472

Dear Reviewers:

We would like to thank the reviewers for careful reading and helpful comments. We revised the manuscript thoroughly according to the comments. The added items are highlighted in red in the main manuscript and the supplementary information. The following changes were made and listed below:

Reviewer #3

Comments 1:

"The authors have duly considered the comments raised by the various reviewers. In addressing my comment on the amount of H₂ and O₂ being generated, I believe the authors accidentally switched the values of H₂ and O₂ in the main text on page 17: "To understand the CER inhibition via the PTh/BVO, the amounts of O₂ and H₂ products during PEC seawater splitting of 10 hours were measured, which are approximately 795.8 and 369.2 μ mol, respectively, (Supplementary Fig. 29)." => There should be more H₂ than O₂, so these values need to switch places."

Response: Thank you for your encouragement and positive comments, which may make our paper more instructive. We have taken your comments seriously and made changes after thinking deeply. The problem you mentioned is our mistake, and the correct expression has been revised in the manuscript.

Comments 2:

"Finally, I would still strongly advise the authors to consult a professional language editor. The level of English employed remains poor. In my review I gave just two examples, representative for the rest of the text. Instead of scrutinizing the entire manuscript as suggested, the authors have addressed these two specific lines, and still even these sentences remain intelligible. For instance: "...photo-induced charges always need across the interface formed by molecular anchoring"? Hence, I recommend acceptance of this manuscript from a scientific perspective, but quite some editing will still be required."

Response: Thank you for your advice. We carefully reviewed the manuscript and corrected some ambiguous and grammatically incorrect sentences and expressions. The sentence "...photo-induced charges always need across the interface formed by molecular anchoring" you mentioned was corrected to "This limitation arises from the necessity of photo-induced charges to traverse across the interface, which is often hindered by unfavorable charge transfer abilities resulting from molecular anchoring." In addition, all revised sentences are shown in the manuscript and highlight in red.